# High Cerebrospinal Fluid CX3CL1 Levels in Alzheimer’s Disease Patients but Not in Non-Alzheimer’s Disease Dementia

**DOI:** 10.3390/jcm11195498

**Published:** 2022-09-20

**Authors:** Giulia Bivona, Matilda Iemmolo, Tommaso Piccoli, Luisa Agnello, Bruna Lo Sasso, Marcello Ciaccio, Giulio Ghersi

**Affiliations:** 1Department of Biomedicine, Neurosciences and Advanced Diagnostics, Institute of Clinical Biochemistry, Clinical Molecular Medicine and Laboratory Medicine, University of Palermo, 90133 Palermo, Italy; 2Department of Laboratory Medicine, University Hospital “P.Giaccone”, 90127 Palermo, Italy; 3Department of Biological, Chemical and Pharmaceutical Sciences and Technologies (STEBICEF), University of Palermo, Viale delle Scienze, Ed. 16, 90128 Palermo, Italy; 4Unit of Neurology, Department of Biomedicine, Neurosciences and Advanced Diagnostics, University of Palermo, 90133 Palermo, Italy

**Keywords:** Alzheimer’s disease, fractalkine, amyloid β, tau

## Abstract

Alzheimer’s disease (AD) is the most common form of cognitive decline worldwide, occurring in about 10% of people older than 65 years. The well-known hallmarks of AD are extracellular aggregates of amyloid β (Aβ) and intracellular neurofibrillary tangles (NFTs) of tau protein. The evidence that Aβ overproduction leads to AD has paved the way for the AD pathogenesis amyloid cascade hypothesis, which proposes that the neuronal damage is sustained by Aβ overproduction. Consistently, AD cerebrospinal fluid (CSF) biomarkers used in clinical practice, including Aβ 1–42, Aβ 1–40, Aβ 42/40 ratio, and pTau, are related to the amyloid hypothesis. Recently, it was suggested that the Aβ deposition cascade cannot fully disclose AD pathogenesis, with other putative players being involved in the pathophysiology of the disease. Among all, one of the most studied factors is inflammation in the brain. Hence, biomarkers of inflammation and microglia activation have also been proposed to identify AD. Among them, CX3 chemokine ligand 1 (CX3CL1) has taken center stage. This transmembrane protein, also known as fractalkine (FKN), is normally expressed in neurons, featuring an N-terminal chemokine domain and an extended mucin-like stalk, following a short intra-cytoplasmatic domain. The molecule exists in both membrane-bound and soluble forms. It is accepted that the soluble and membrane-bound forms of FKN evoke differential signaling within the CNS. Given the link between CX3XL1 and microglial activation, it has been suggested that CX3CL1 signaling disruption could play a part in the pathogenesis of AD. Furthermore, a role for chemokine as a biomarker has been proposed. However, the findings collected are controversial. The current study aimed to describe the cerebrospinal fluid (CSF) levels of CX3XL1 and classical biomarkers in AD patients.

## 1. Introduction

Alzheimer’s disease (AD) is the most common form of cognitive decline worldwide, occurring in about 10% of people older than 65 years [1,2]. AD patients experience loss of memory and cognitive function impairment due to severe and progressive neuronal damage. 

The well-known hallmarks of AD are extracellular aggregates of amyloid β (Aβ) and intracellular neurofibrillary tangles (NFTs) of tau protein. The evidence that Aβ overproduction leads to AD has paved the way for the AD pathogenesis amyloid cascade hypothesis, which proposes that the neuronal damage is sustained by Aβ overproduction. However, it has been suggested that the amyloid cascade hypothesis cannot fully disclose AD pathogenesis [3]. Hence, different processes, separate from and interacting with the amyloid cascade, have been deemed to be involved in AD pathophysiology; one of the most studied is inflammation in the brain [4,5,6,7]. Termed neuroinflammation, this phenomenon relies on the activation of microglia and is related to many neurodegenerative diseases [2,7]. Evidence supporting a link between microglial activation and AD pathogenesis includes the following observations: (i) a correlation between plaque burden and microglial activation in mouse models; (ii) activated microglia surrounding plaques; (iii) high proinflammatory marker levels in AD patients; (iv) persistent activation of microglia during AD even when the plaque burden decreases, indicating chronic activation of microglia; (v) correlation of AD risk genes with native immunity [2,7,8]. Theoretically, activation of microglia is a transient and protective mechanism, occurring after a pathogenic stimulus has evoked a response for the first time. It is accompanied by the release of anti-inflammatory molecules to avoid sustained inflammation, which can lead to neuronal dysfunction and loss [7]. However, some physiological and pathological conditions, including aging and baseline inflammation, can transform microglia toward a less protective phenotype, resulting in a more persistent and aggressive response to injury. This represents the so-called “priming” of the microglia and occurs upon repeated stimulation [2]. 

Neuroinflammatory mechanisms may differ between AD and other tauopathies [9,10,11]. CX3 chemokine ligand 1 (CX3CL1), also known as fractalkine (FKN), is a transmembrane protein normally expressed in neurons featuring an N-terminal chemokine domain and an extended mucin-like stalk, following a short intra-cytoplasmatic domain [2,12]. Throughout the central nervous system (CNS), FKN is mainly expressed in the hippocampus [13], where it interacts with its receptor CX3CR1, which is expressed by microglia and neurons [14]. CX3CL1 reduces microglial activation and inhibits proinflammatory gene expression and cytokine (IL-1 β, IL-6, and TNF-α) synthesis [15]. This maintains the hippocampus microenvironment in a quiescent/anti-inflammatory state, which is fundamental for neuronal progenitor cells (NPSs) of the neurogenic niche to drive neurogenesis [16,17,18].

Some stimuli and conditions, including Aβ extracellular accumulation and activation of inflammation, can modify CX3CL1/CX3CR1 signaling [7]. Notably, it has been reported that, during AD, some changes in the chemokine activity can occur according to the stage of the disease [19]. Furthermore, it has been suggested that dysfunctional FKN signaling could favor microglia priming after activation [2]. These data have led to a hypothetical role for this chemokine as a biomarker. Classical CSF biomarkers for AD include Aβ 1–42, Aβ 1–40, Aβ 42/40 ratio, and pTau, and their clinical usefulness for identifying the disease is well established. Conversely, little evidence exists on the role of FKN as a diagnostic tool in AD, mainly because studies measuring CSF levels of CX3XL1 in AD patients are lacking compared to those on classical biomarkers.

The current study was aimed at describing the CSF of CX3XL1 in AD patients along with classical biomarkers.

## 2. Materials and Methods

### 2.1. Selection of Patients and Groups

For this study, we selected 28 patients affected by AD and a control group of 18 subjects affected by cognitive decline not related to AD. AD patients were diagnosed according to current criteria (Albert, 2011; MacKahnn, 2011) [20,21]. All patients were recruited from the Clinic for Cognitive Decline, Dementia, and Parkinsonism of the University Hospital Paolo Giaccone Palermo, Italy and underwent a general and neurological examination, cognitive evaluation, brain MRI, and FDG-PET, as well as a lumbar puncture, during the diagnostic work-up. Then, all patients were classified according to the AT (N) biomarker classification (Clifford R. Jack, Jr., 2018) [22]. Group A (mean age 70 ± 8; F/M = 0.75) consisted of patients categorized as “Alzheimer’s continuum”, containing only A+ T± (N+) patients; Group B (*n* = 18; mean age 66.7 ± 10; F/M = 0.83) consisted of A− T− (N+) patients (i.e., “non-AD pathological changes”). Inclusion criteria for Group A were as follows: diagnosis of mild cognitive impairment due to AD (Albert, 2011) [20] or probable AD dementia with evidence of AD pathophysiological process (McKhann, 2011) [21], as well as being part of “Alzheimer’s continuum” according to AT (N) classification. Exclusion criteria for Group A were as follows: any other medical condition explaining cognitive decline, including other degenerative diseases, cerebrospinal disease, metabolic disease, and substance abuse. CSF was collected during morning hours in polypropylene tubes, centrifuged at 2000 rpm for 20 min, and stored at −80 °C until analysis [23]. All patients gave their written informed consent, and all procedures were conducted in accordance with the Declaration of Helsinki. The study protocol was approved by the EC of University Hospital “Paolo Giaccone Palermo” (Institutional Ethic Commettee Palermo1 No. 07/2017). 

### 2.2. Evaluation of Aβ 1–40, Aβ 1–4, Tau-Total and Tau-Phosphorylated Using Chemiluminescence Enzyme Immunoassay (CLEIA) in Selected Groups

CSF CLEIA evaluation of all the patients under examination for the concentration of amyloid fibers, Aβ 1–40 and Aβ 1–42, and tau protein, both the hyperphosphorylated form (Tau-phosphorylated) and the total (Tau-total), was performed. In particular, we used the following markers in the identification of different molecules: Lumipulse G β-amyloid 1–40, Lumipulse G β-amyloid 1–42, Lumipulse G Total Tau, and Lumipulse Gp Tau 181 (in hyperphosphorylated Tau identification), from Fujirebio Inc. Erope, Gent, Belgium on a fully automated platform (Lumipulse G1200 analyzer Fujirebio Inc. Erope, Gent, Belgium) according to the manufacturer’s instructions.

### 2.3. CX3CL1 Evaluation Using Enzyme Linked Immunosorbent Assay (ELISA) in CSF Patients

To evaluate the amount of human fractalkine (chemokine CX3CL1, soluble form) in the CSF of analyzed patients, a sandwich enzyme-linked immunosorbent assay was applied. In particular, the Wuhan Fibe Biotech Co., Ltd. (Wuhan, China) was used, according to the manufacturer’s instructions. Briefly, the CSF obtained from patients (as previously described in Section 2.1) was converted into liquid form by cold thawing and then returned to room temperature, before mixing 1:1 in dilution buffer (f.v. 100 μL). Samples were incubated together with standards and background controls for 90 min at 37 °C, washed and incubated with biotin-labeled antibody for 60 min at 37 °C, and then washed and incubated with 3,3′,5,5′-tetramethylbenzidine (TMB) for 10–20 min at 37 °C. Colorimetric reactions were stopped by adding Stop Solution and analyzed using a spectrophotometer (microplate reader DU-730 Life Science spectrophotometer (Beckman Coulter, Milan, Italy)) at an OD of 450.

### 2.4. Statistical Analyses

The data obtained were evaluated for normality by applying the Shapiro–Wilk test. The variance found between samples, as a function of the analyzed variables, was determined according to one-way ANOVA (*p*-value).

Given the limited number of subjects analyzed (a canonical Student’s *t*-test was not applicable) for the presence of different molecules in the CSF, for the statistical analysis of the data, we applied the Mann–Whitney U test (nonparametric test). Specifically, the ratio of the mean of the medians between non-AD and AD subjects was evaluated. The null hypothesis (H0—equality of the values of the two groups) was rejected for *p* < 0.05.

## 3. Results

### 3.1. Patient Classification into Groups

The recruited patients, as reported in Section 2, were divided into two groups according to the AT (N) biomarker classification: “Alzheimer’s continuum“ (Group A) and “non-AD pathological changes” (Group B). Group B comprised seven patients with vascular dementia (38.8%), two patients with Lewy bodies (11.1%), four patients with frontotemporal dementia (22.2%), one patient with Parkinson’s disease (5.5%), one patient with subjective cognitive decline (5.5%), two patients with corticobasal degeneration (11.1%), and one patient with paraneoplastic syndrome (5.5%).

As shown in Figure 1, the CSF of the subjects was evaluated by CLEIA for the quantity of Aβ 1–42 (Figure 1A) and Aβ 1–40 (Figure 1B), as well as the Aβ 42/40 ratio (Figure 1C). Moreover, the expression of tau, as illustrated in Figure 2, was quantified in terms of Tau-total (Figure 2A) and Tau-phosphorylated (Figure 2B). In Figure 1 and Figure 2, the *p*-values report the significance of differences in the population comparison. Table 1 and Table 2 show the average values of the different variables, together with the *p*-value (one-way ANOVA), in the two groups of subjects with respect to the total population to determine the non-randomness of the values obtained. In support of the results, Appendix A show the values measured in the CSF for all patients in terms of the following variables: Aβ 1–42 (Appendix A), Aβ 1–40 (Appendix A), Aβ 42/40 ratio (Appendix A), Tau-total (Appendix A), and Tau-phosphorylated (Appendix A). Therefore, the two groups were determined as follows: Group A included subjects who showed values outside the norm for all variables; Group B (control group) included healthy subjects in which all values of the analyzed variables fell within the norm. 

However, as shown in Appendix A, the expression analysis of Tau-total and Tau-phosphorylated for some patients belonging to Group A showed values closer to non-AD subjects (Group B), even if the clinical parameters and conditions were reflective of AD. For a better comparison regarding the presence of CX3CL1 in the cerebrospinal fluid of all analyzed subjects, we split group A into two subgroups according to AT (N) classification as follows: Group A’ (16 patients; A1–A16; A+ T+ N+) and Group A” (12 patients; A17–A28; A+ T− N+). The expression of Aβ 1–42 and Aβ 1–40, and the Aβ 42/40 ratio in the CSF are shown in Figure 3 and summarized in Table 3; both subpopulations (A’ and A”) had Aβ values canonically attributable to AD patients.

Table 3 shows the average values of the different variables, together with their *p*-value (one-way ANOVA), for a comparison of Groups A’, A”, and B. In support of these data, Appendix A show the values measured for all analyzed subjects (Groups A’, A”, and B) in terms of Aβ 1–42 (Appendix A), Aβ 1–40 (Appendix A), and the Aβ 42/40 ratio (Appendix A).

The behavior differed as a function of Tau-total and Tau-phosphorylated. As shown in Figure 4 and Table 4, the population of Group A’ showed values ascribable to AD for both markers (Aβ, Tau-total, and Tau-phosphorylated); on the other hand, in Group A”, the values of Tau-total and Tau-phosphorylated were within normal limits, i.e., similar to patients of Group B, the non-AD subjects.

Table 4 shows the average values of the different variables, together with their *p*-value (one-way ANOVA), for a comparison of Groups A’, A”, and B. In support, Appendix A show the values measured for all subjects analyzed in terms of Tau-total (Appendix A and Tau-phosphorylated (Appendix A).

### 3.2. CX3CL1 Expression in the Different Patient Groups

The CSF of patients belonging to Groups A’, A”, and B (classified as decribed in Section 2) was compared using ELISA for the presence of CX3CL1. As shown in Figure 5, the level of CX3CL1 differed quantitatively, with Group A’ and Group A” having 33% and 40% greater levels of CX3CL1, respectively, compared to Group B. Table 5 shows a comparison the average values of CX3CL1 for each group together with the σ^2^ value (one-way ANOVA).

Appendix A reports the amount of CX3CL1 in the CSF of each patient.

According to the analysis of the values in Groups A and B, Figure 6 reveals that CX3CL1 was present in the CSF of all subjects; however, Group A had a 36% greater level of CX3CL1 compared to Group B.

Furthermore, Table 6 compares the average values of CX3CL1 for each group together with the *p*-value (one-way ANOVA). Appendix A reports the amount of CX3CL1 in the CSF of patients in Group A and Group B.

### 3.3. Statistical Evaluation According to Mann–Whitney Test

Table 7 reports the statistical analysis of the results according to the Mann–Whitney U-test, as described in Section 2. It is suggested that the null hypothesis could be rejected in most cases.

## 4. Discussion

The current evaluation of CX3CL1 levels in the CSF of AD patients is one arm of a larger study investigating the role of CX3CL1/CX3CR1 signaling in the pathogenesis of AD. In vitro (neuron–glia coculture) and in vivo (AD rat models) systems were used to evaluate the changes in the intracellular expression and localization of CX3CL1, as well as some transcription factors involved in its signaling cascade, including p38 and β catenin. The project’s final goal is to identify possible targets for novel treatment strategies among the molecules involved in the CX3CL1 pathway, including many transcription factors and metalloproteinases. To reach this goal, it is mandatory to evaluate the activation (or response) of the microglia with respect to various stimuli, including Aβ and Tau protein, considering their different effects. For instance, Aβ fibrils vs. Aβ oligomers and Aβ aggregates vs. tau aggregates elicit diverse activation mechanisms in the microglia, resulting in different phenotypes or subsets of these cells. It is well known that some phenotypes and mechanisms, i.e., “primed” microglia, are strongly associated with the progression of the disease, representing a critical element within the pathophysiology of AD [2]. In the current study, CSF specimens were investigated for the different AT (N) characteristics with the aim of evaluating possible responses evoked by the microglia. Significant changes in the expression of transcription factors in the in vivo system were found, being overexpressed in the coculture for group A’ patients (data not shown). Differences in Groups A’ and A” according to the AT (N) classification reveal the extent to which Aβ and pTau can elicit different response in the microglia within the coculture and AD rat model.

In the present study, high CX3CL1 concentrations were found in the CSF of AD patients compared to non-AD patients. Similar results in other studies have been achieved, although a few analyses reported opposite findings. For example, Nordengen et al. [24] performed an analysis of 61 AD patients divided into four subgroups according to the AT (N) classification: A+ T+ (N+), A+ T− (N+), A+ T− (N−), and A− T− (N−). The authors found significantly higher CX3CL1 levels in the CSF of A+ T+ (N+) and A+ T− (N+) groups compared to subjects in the A+ T− (N−) group and healthy controls (A− T− (N−)) [21]. Despite the similarity in the results between Nordengen et al. and the present study, it should be noted that, in the current analysis, A− T− (N+) subjects were considered as controls (non-Alzheimer’s pathophysiology (SNAP) according to the NIA-AA classification) [22], whereas, in Nordengen et al.’s study, they were included among the patients (MCI group and subjective cognitive decline (SCD) group), instead using the A− T− (N−) CSF pattern to represent the healthy controls. 

Kulczynska-Przybik et al. [25] also found similar findings when performing CX3CL1 measurements in the CSF of 42 AD and 18 MCI patients. The authors found CX3CL1 to be higher in MCI and AD patients as compared to the healthy controls. However, the study group was not categorized according to the NIA-AA biomarker classification, making it difficult to compare the results with the current study [22].

Van Ton et al. [26] recently evaluated the CSF expression of many proteins, including FKN, in AD patients, reporting that the chemokine is upregulated in AD patients. Despite the similarity in the results obtained, it is worth noting that Van Ton et al. used a proximity extension assay (PEA) to measure CSF protein levels, whose analytical features (sensitivity, specificity, and accuracy) are distinct from those of the methods used in the current study [23]. Furthermore, Van Ton et al.’s selection of patients was slightly different from that in the present study, since cognitive decline was defined according to the Clinical Dementia Rating Scale (CDRS) and the patient group included both mild cognitive impairment (MCI) and AD patients. 

Perea et al. [27] achieved opposite findings, reporting decreased CX3CL1 levels in the CSF of AD patients. Once more, the patients were not comparable as Perea et al. did not use the AT (N) biomarker pattern classification to define MCI and AD patients. Furthermore, the sample size was relatively small (42 CSF specimens in total from healthy controls, MCI subjects, and AD subjects). 

The increase in CX3CL1 in the CSF of AD patients is not surprising. The influence of CX3CL1 on Aβ clearance and Tau phosphorylation is well established. Briefly, CX3CL1 typically inhibits tau phosphorylation, but promotes Aβ accumulation, since Aβ clearance requires a proinflammatory behavior of the microglia, whereas CX3CL1 signaling maintains the microglia in a resting state [15]. Hence, the disruption of CX3CL1 signaling results in enhanced Tau phosphorylation and reduced Aβ accumulation. It has been hypothesized that the accumulation of Aβ at the very early stage of disease results in dysfunctional CX3CL1/CX3CR1 signaling, thus increasing Tau phosphorylation and Aβ clearance by the microglia [28].

A significant issue regards the difference in CX3CL1 levels in the CSF of subjects with and without neurodegeneration, as defined by the NIA-AA criteria. Given that neurodegeneration pathophysiology features an inflammatory basis, it is reasonable to ask why this chemokine does not show an increase in patients with SNAP. However, the role of CX3CL1 in microglial activation, along with its contradictory Janus behavior with respect to Aβ accumulation and Tau phosphorylation, could explain why AD but not SNAP patients showed altered levels of CX3CL1 in the CSF.

## 5. Limitations

The current study had some limitations, such as its observational design, small sample size, and single-center nature, as well as the absence of plasma level measurements of chemokine, and the absence of in vivo and in vitro analysis results (ongoing) supporting and explaining the present findings.

## 6. Conclusions

AD is the most common form of dementia, representing a major health burden worldwide. The diagnosis of AD benefits from the biochemical measurement of CSF proteins that are required to define AD. However, determination of these biomarkers is time-consuming and expensive, requires analytical methods and specialized equipment, and is not yet adapted for point-of-care (POC) diagnostics. Novel potential biomarkers to identify patients within the AD continuum include inflammatory markers and chemokines, due to the undeniable role of neuroinflammation and microglia activation in AD pathogenesis. The research on this topic has recently focused on a specific target to be studied for clinical and therapeutic purposes, i.e., the CX3CL1/CX3CR1 axis, whose signaling maintains the microglia in a resting state and favors the formation of memories within the hippocampus. The main goal is the identification of biomarkers with good diagnostic and prognostic value, which undergo changes on the basis of the AD continuum in the preclinical, prodromal, and dementia phases. Lastly, it is worth mentioning that extensive evidence supports the role of CX3CL1/CX3CR1 axis disruption in neuronal damage leading to cognitive impairment and dementia in AD patients. Hints for future research in this field include the identification of molecules (ligands, receptors, and transcription factors) within the CX3CL1/CX3CR1 signaling pathway that can be used as targets for treatment strategies. FKN signaling defines the baseline inflammation in the brain, which influences how microglia shift toward different phenotypes. CX3CL1/CX3CR1 dysfunction could favor microglia priming, which is a key element making the brain microenvironment more prone to developing chronic neuroinflammation. Targeting molecules of the CX3CL1/CX3CR1 signaling pathway could help in the discovery novel treatment strategies for AD.

## Figures and Tables

**Figure 1 jcm-11-05498-f001:**
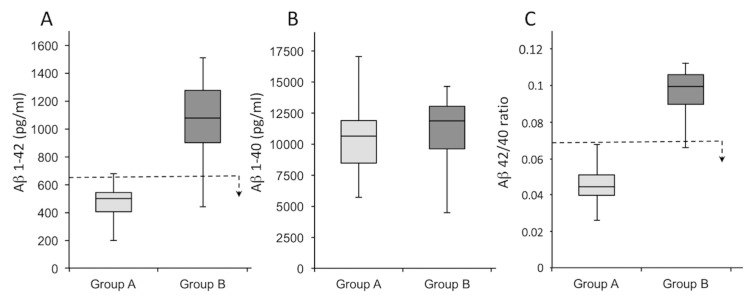
Subdivision of patients into two groups regarding the expression of Aβ 1–42 (**A**) and Aβ 1–40 (**B**), and the Aβ 42/40 ratio (**C**). The dotted line represents the threshold for a subject to be considered AD or non-AD; the arrow indicates the direction of values for which the subjects are affected by AD. Group A is depicted in light gray, and Group B is depicted in dark gray.

**Figure 2 jcm-11-05498-f002:**
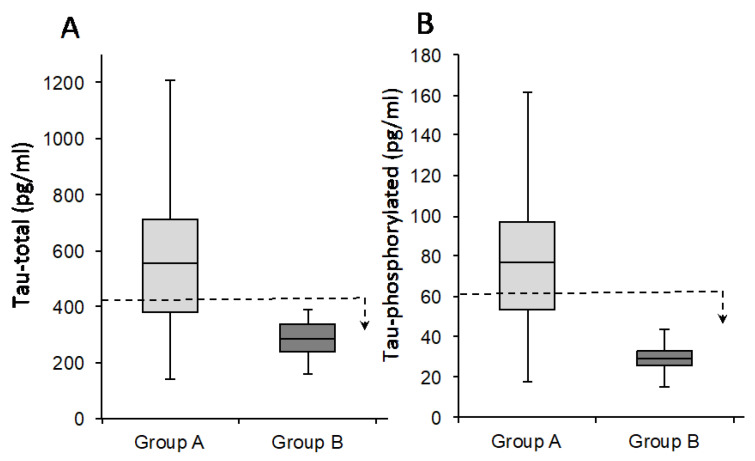
Subdivision of patients into two groups regarding the expression of Tau-total (**A**) and Tau-phosphorylated (**B**). The dotted line represents the threshold for a subject to be considered AD or normal; the arrow indicates the direction of values for which the subjects are not affected by AD. All groups were evaluated for significance according to Student’s *t*-test (*p* < 0.01). Group A is depicted in light gray, and Group B is depicted in dark gray.

**Figure 3 jcm-11-05498-f003:**
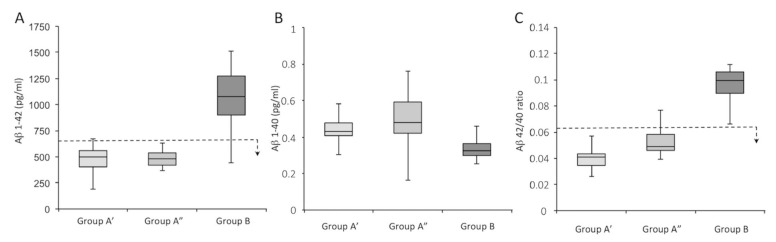
Subdivision of patients into Groups A’, A”, and B according to the expression of Aβ 1–42 (**A**) and Aβ 1–40 (**B**), and the Aβ 42/40 ratio (**C**). The dotted line represents the threshold for a subject to be considered AD or normal; the arrow indicates the direction of values for which the subjects are affected by AD. Group A’ is depicted in light gray, Group A” is depicted in medium gray, and Group B is depicted in dark gray.

**Figure 4 jcm-11-05498-f004:**
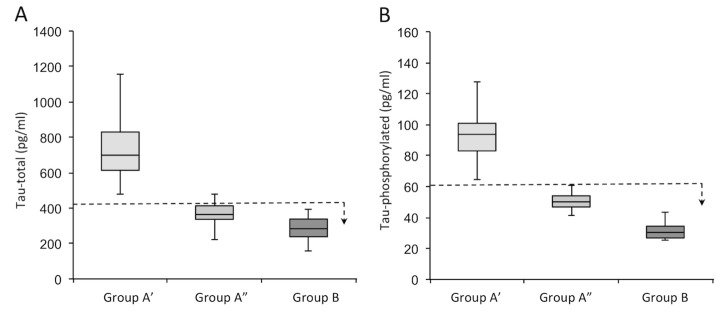
Subdivision of patients into Groups A’, A”, and B according to the expression of Tau-total (**A**) and Tau-phosphorylated (**B**). The dotted line represents the threshold for a subject to be considered AD or normal; the arrow indicates the direction of values for which the subjects are not affected by AD. Group A’ is depicted in light gray, Group A” is depicted in medium gray, and Group B is depicted in dark gray.

**Figure 5 jcm-11-05498-f005:**
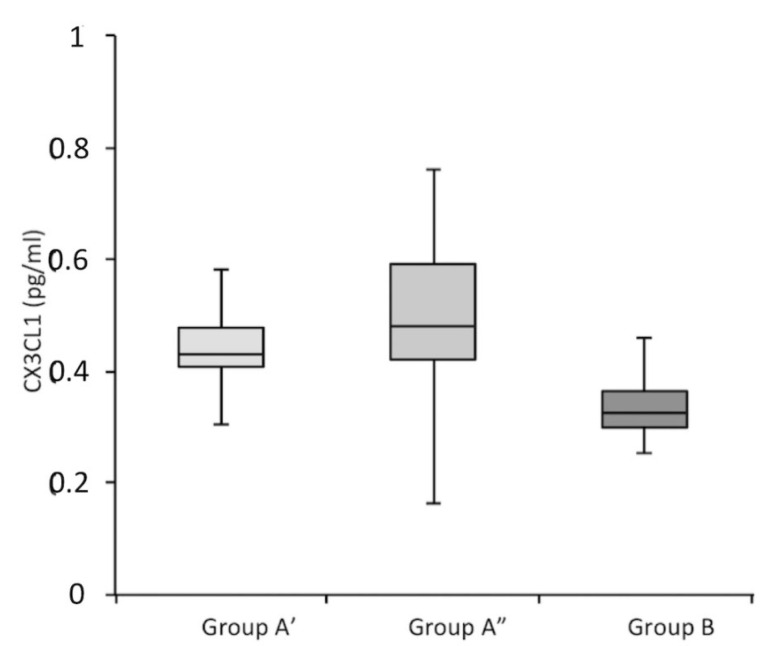
CX3CL1 in the CSF of subjects belonging to Groups A’, A”, and B. Group A’ is depicted in light gray, Group A” is depicted in medium gray, and Group B is depicted in dark gray.

**Figure 6 jcm-11-05498-f006:**
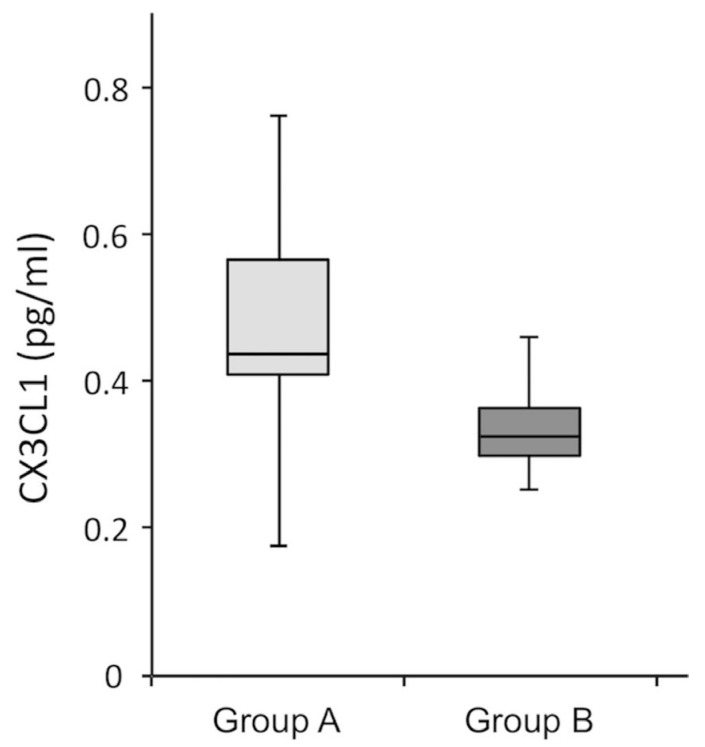
CX3CL1 in the CSF of subjects belonging to Groups A and B. The significance of the differences was evaluated using Student’s *t*-test (*p* < 0.01). Group A is depicted in light gray, and Group B is depicted in dark gray.

**Table 1 jcm-11-05498-t001:** Average values recorded for each variable. The minimum and maximum values are shown in parentheses; σ^2^ is the variance of the values according to “one-way ANOVA”.

Variable	Patients Tot (*n* = 46)	Group A (*n* = 28)	Group B (*n* = 18)	σ^2^ *
Aβ 1–42 (pg/mL)	707 (190–1512)	479 (190–676)	1063 (440–1512)	0.78
Aβ 1–40 (pg/mL)	10,877 (4407–19,441)	11,840 (7184–19,441)	11,086 (4407–14,490)	0.05
Aβ 42/40 ratio	0.07 (0.03–0.011)	0.04 (0.03–0.06)	0.10 (0.07–0.11)	0.03

* One-way ANOVA.

**Table 2 jcm-11-05498-t002:** Average values recorded for each variable. The minimum and maximum values are shown in parentheses; σ^2^ is the variance of the values according to “one-way ANOVA”.

Variable	Patients Tot (*n* = 46)	Group A (*n* = 28)	Group B (n = 18)	σ^2^ *
Tau-total	470 (140–1561)	770 (477–1561)	278 (173–390)	0.45
Tau-phossphorilated (181)	63 (11–232)	111 (64–232)	29 (11–43)	0.71

* One-way ANOVA.

**Table 3 jcm-11-05498-t003:** Average values recorded for each variable, Aβ 1–42, Aβ 1–40 and Aβ 42/40 ratio. The minimum and maximum values are shown in parentheses; σ^2^ is the variance of the values according to “one-way ANOVA”.

Variable	Patients Tot (*n* = 46)	Group A’ (*n* = 16)	Group A” (*n* = 12)	Group B (*n* = 18)	σ^2^ *
Aβ 1–42 (pg/mL)	707 (190–1512)	479 (190–676)	480 (378–630)	1063 (440–1512)	0.56
Aβ 1–40 (pg/mL)	10,877 (4407–19,441)	11,840 (7184–19,441)	9279 (5698–11,716)	11,086 (4407–14,490)	0.29
Aβ 42/40 ratio	0.07 (0.03–0.011)	0.04 (0.03–0.06)	0.05 (0.04–0.09)	0.10 (0.07–0.11)	0.52

* One-way ANOVA.

**Table 4 jcm-11-05498-t004:** Average values recorded for each variable, Tau-total and Tau-phosphorilated. The minimum and maximum values are shown in parentheses; σ^2^ is the variance of the values according to “one-way ANOVA”.

Variable	Patients Tot (*n* = 46)	Group A’ (*n* = 16)	Group A” (*n* = 12)	Group B (*n* = 18)	σ^2^ *
Tau-total	470 (140–1561)	770 (477–1561)	359 (140–473)	278 (173–390)	0.67
Tau-phosphorilated (181)	63 (11–232)	111 (64–232)	50 (18–63)	29 (11–43)	0.60

* One-way ANOVA.

**Table 5 jcm-11-05498-t005:** Average values recorded for CX3CL1. The minimum and maximum values are shown in parentheses; σ^2^ is the variance of the values according to “one-way ANOVA”.

Variable	Patients Tot (*n* = 46)	Group A’ (*n* = 16)	Group A” (*n* = 12)	Group B (*n* = 18)	σ^2^ *
CX3CL1	0.43 (0.25–0.76) ± 0.15	0.47 (0.28–0.75) ± 0.03	0.50 (0.13–0.76) ± 0.03	0.35 (0.25–0.74) ± 0.04	0.46

* One-way ANOVA.

**Table 6 jcm-11-05498-t006:** Average values recorded for CX3CL1. The minimum and maximum values are shown in parentheses; σ^2^ is the variance of the values according to “one-way ANOVA”.

Variable	Patients Tot (*n* = 46)	Group A (*n* = 28)	Group B (*n* = 18)	σ^2^ *
CX3CL1	0.43 (0.25–0.76) ± 0.15	0.47 (0.28–0.75) ± 0.03	0.35 (0.25–0.74) ± 0.04	0.01

* One-way ANOVA.

**Table 7 jcm-11-05498-t007:** Mann-Whitney medians’ media evaluation test. The ratio of the mean of the medians of Group B respect the values of each Group A (A, A’ and A”) was evaluated together with the *p*-value. For the hypothesis H0 = Null hypothesis (equality of the values of the two analyzed groups) this was rejected for *p* < 0.05.

		Group A	Group A’	Group A”	Group B
Aβ 1–42	Median value	508.50	573.98	477.33	1064.5
	Median value GroupB/Median group value	2.09	1.87	2.23	
	*p*-value	4.09 × 10^−9^	3.0 × 10^−5^	2.24 × 10^−4^	
	*H0	RH0	RH0	RH0	
Aβ 1 = −40	Median value	13,539	14,319	9611.8	11,669
	Median value GroupB/Median group value	0.86	0.81	1.21	
	*p*-value	5.26 × 10^−2^	4.60 × 10^−1^	2.23 × 10^−1^	
	*H0	NRH0	NRH0	NRH0	
Aβ 42/40 ratio	Median value	0.0364	0.0408	0.0532	0.0909
	Median value GroupB/Median group value	2.49	2.23	1.71	
	*p*-value	4.88 × 10^−2^	1.84 × 10^−2^	2.39 × 10^−2^	
	*H0	RH0	RH0	RH0	
Tau-total	Median value	893.48	752.2	313.59	312.66
	Median value GroupB/Median group value	0.35	0.42	0.99	
	*p*-value	1.05 × 10^−5^	1.24 × 10^−9^	5.6 × 10^−3^	
	*H0	RH0	RH0	RH0	
Tau- phosphorilated	Median value	125.64	94.373	48.674	32.879
	Median value GroupB/Median group value	0.26	0.34	0.67	
	*p*-value	2.23 × 10^−6^	2.26 × 10^−9^	2.19 × 10^−6^	
	*H0	RH0	RH0	RHO	
CX3CL1	Median value	0.4789	0.5568	0.4326	0.3195
	Median value GroupB/Median group value	0.67	0.57	0.74	
	*p*-value	1.05 × 10^−2^	3.75 × 10^−2^	6.75 × 10^−1^	
	*H0	RH0	RH0	NRH0	

*H0 = Null hypothesis; RH0 = Reject null hypothesis; NRH0 = Don’t reject null hypothesis.

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
