# Peer review of "High Cerebrospinal Fluid CX3CL1 Levels in Alzheimer’s Disease Patients but Not in Non-Alzheimer’s Disease Dementia"

_jcm, 2022, doi:10.3390/jcm11195498_

Round 1
Reviewer 1 Report
The issue concerning pathophysiological features. of Alzheimer's Disease (AD) is vital.
I have several comments, which should be addressed before further consideration:
1. In the introduction authors present an overview on AD. It would valuable to mention that AD may be a clinical manifestation, as well as pathology with different clinical manifestations as Corticobasal Syndrome (CBS) - Ref.
Difficulties in the diagnosis of four repeats (4R) tauopathic parkinsonian syndromes. Neurol Neurochir Pol. 2018 Aug;52(4):459-464. doi: 10.1016/j.pjnns.2018.06.002. Epub 2018 Jul 3. PMID: 30025721.
2. Authors state that:
"Also named neuroinflammation, 48 this phenomenon relies on the activation of microglia and has been related to many neu- 49 rodegenerative diseases"
It would be worth mentioning how the neuroinflammatory mechanism differs in AD and other tauopathies.
Ref.
Neuroinflammation predicts disease progression in progressive supranuclear palsy. J Neurol Neurosurg Psychiatry. 2021 Jul;92(7):769-775. doi: 10.1136/jnnp-2020-325549. Epub 2021 Mar 17. PMID: 33731439; PMCID: PMC7611006.
Neutrophil-to-lymphocyte ratio (NLR) at boundaries of Progressive Supranuclear Palsy Syndrome (PSPS) and Corticobasal Syndrome (CBS). Neurol Neurochir Pol. 2021;55(1):97-101. doi: 10.5603/PJNNS.a2020.0097. Epub 2020 Dec 14. PMID: 33315235.
It would be valuable to expand this issue in the discussion.
3. Authors should elaborate on future perspectives.
Author Response
Comments and Suggestions for Authors
The issue concerning pathophysiological features. of Alzheimer's Disease (AD) is vital.
I have several comments, which should be addressed before further consideration:
- In the introduction authors present an overview on AD. It would valuable to mention that AD may be a clinical manifestation, as well as pathology with different clinical manifestations as Corticobasal Syndrome (CBS) - Ref. Difficulties in the diagnosis of four repeats (4R) tauopathic parkinsonian syndromes. Neurol Neurochir Pol. 2018 Aug;52(4):459-464. doi: 10.1016/j.pjnns.2018.06.002. Epub 2018 Jul 3. PMID: 30025721.
1) The suggestion has been addressed, and the reference has been added.
- Authors state that
"Also named neuroinflammation, 48 this phenomenon relies on the activation of microglia and has been related to many neu- 49 rodegenerative diseases". It would be worth mentioning how the neuroinflammatory mechanism differs in AD and other tauopathies. Ref: Neuroinflammation predicts disease progression in progressive supranuclear palsy. J Neurol Neurosurg Psychiatry. 2021 Jul;92(7):769-775. doi: 10.1136/jnnp-2020-325549. Epub 2021 Mar 17. PMID: 33731439; PMCID: PMC7611006. Ref: Neutrophil-to-lymphocyte ratio (NLR) at boundaries of Progressive Supranuclear Palsy Syndrome (PSPS) and Corticobasal Syndrome (CBS). Neurol Neurochir Pol. 2021;55(1):97-101. doi: 10.5603/PJNNS.a2020.0097. Epub 2020 Dec 14. PMID: 33315235.
It would be valuable to expand this issue in the discussion.
2) The suggestion has been addressed, and the references have been added.
- Authors should elaborate on future perspectives
3) Future perspectives have been addressed more extensively in the paragraph “Conclusions”.

Reviewer 2 Report
From the initial abstract of the manuscript, I was very excited to read the findings and quite interested to know what the authors discover about the role of CX3CL1 levels detected in the CSF of Alzheimer's disease patients compare to controls and whether there is a basis of using it as an additional biomarker for diagnosing AD.
However, upon reading the manuscript in detail, I was honestly disappointed in the way the data is presented and I honestly feel that the results of this manuscript can be presented much better and clearer for the reader to appreciate the findings. Even if the sample size of the study is relatively small, 28 patients in Group A and 18 patients in Group B, the authors should be able to clearly define what tests were performed, the results and how should one interpret them. Below are my comments,
1) The authors emphasized a lot on CX3CL1 in the title and abstract. However, in the body of the manuscript, almost all the data were about testing Aβ42, Aβ40, Aβ42/Aβ40 ratio and pTau. The CX3CL1 data is only presented in the last figure and it is not clear to me after reading the entire manuscript why is CX3CL1 the focus of the manuscript? It would seem to be more logical for the authors to present the data as a logical test of Aβ42, Aβ40, Aβ42/Aβ40 ratio, pTau and CX3CL1 between the patient groups.
2) As for patient groups, while the authors did define what Group A and Group B are, i.e Group A are patients on the AD continuum and Group B are "non-AD" patients, I was unable to determine what was the logical difference between Group A' and Group A''. This is important given that several analyses (Figure 3 onwards) were performed with A' as a group independent of A''.
3) The reported statistics of the results is not coherent. From the methods (2.4. Statistical Analyses.),
"While the variance found between the different samples as a function of the analyzed variables was determined according to one-way ANOVA (p* value). Furthermore, through the distribution according to student' T, the different samples for the different variables were compared about the significance of the data obtained (p-value)."
I am sorry to say, but the above is not coherent. Am I suppose to assume that it is a one-way ANOVA test (for those with Group A', Group A'' and Group B) and the P-values were reported (which is labeled as p* value)? If so, why is the Aβ 42/40 ratio P-value result in Figure 1 shown as 1.61? A P-value as commonly understood should not have values above 1. The manuscript says that p* value is 'the variance of the values', which is not the case as the vast majority of the time, σ2 is the variance, not p*.
3) Figure 1 as an image is not coherently depicted either in the manuscript.
The above is a screen capture of what I see and it seems that the figure was badly resized and also flipped. There are also no legends detailing which one is Group A and which one is Group B. The caption text, "All groups were valuate in significance about T of student shown a p < 0.001." is also incoherent. Is this suppose to be a Student's T-test?
4) In Figure 5, I am guessing that the P-values were depicted on the Figure based on performing a Student's T-test between the 2 of the Groups, i.e. Group A' vs Group A'', Group A' vs Group B, Group A'' vs Group B. However, the P-values does not seem to match the box-plots. I would assume the Group A' vs Group B and Group A'' vs Group B comparisons should have a more significant P-value than the Group A' vs Group A'' comparison. From just looking at the box-plots, it is hard for me to believe that the P-value of Group A' vs Group A'' is significant at P<0.001.
Overall, I think there is worthy data in the manuscript to report. However, I would suggest that the authors edit the manuscript extensively to better present the data, analyses and results.
Author Response
Comments and Suggestions for Authors
From the initial abstract of the manuscript, I was very excited to read the findings and quite interested to know what the authors discover about the role of CX3CL1 levels detected in the CSF of Alzheimer's disease patients compare to controls and whether there is a basis of using it as an additional biomarker for diagnosing AD.
However, upon reading the manuscript in detail, I was honestly disappointed in the way the data is presented and I honestly feel that the results of this manuscript can be presented much better and clearer for the reader to appreciate the findings. Even if the sample size of the study is relatively small, 28 patients in Group A and 18 patients in Group B, the authors should be able to clearly define what tests were performed, the results and how should one interpret them. Below are my comments,
1) The authors emphasized a lot on CX3CL1 in the title and abstract. However, in the body of the manuscript, almost all the data were about testing Aβ42, Aβ40, Aβ42/Aβ40 ratio and pTau. The CX3CL1 data is only presented in the last figure and it is not clear to me after reading the entire manuscript why is CX3CL1 the focus of the manuscript? It would seem to be more logical for the authors to present the data as a logical test of Aβ42, Aβ40, Aβ42/Aβ40 ratio, pTau and CX3CL1 between the patient groups.
1) We agree with the Reviewer's comment and, consistently, the abstract, title and introduction have been changed, to shorten the dissertation on CX3CL1 and highlight that the paper reports findings on both classical biomarkers and CX3CL1 CSF levels.
2) As for patient groups, while the authors did define what Group A and Group B are, i.e Group A are patients on the AD continuum and Group B are "non-AD" patients, I was unable to determine what was the logical difference between Group A' and Group A''. This is important given that several analyses (Figure 3 onwards) were performed with A' as a group independent of A''.
2) We agree with the Reviewer, and we have provided within the text, in the section Discussion, a brief explanation for the categorisation of patients in A' and A" to make the Readers understand the logical sense of such a subdivision. A much more detailed explanation to the Reviewer follows:
In the aim to measure CX3CL1 in AD patients and compare it to a non-AD control group, patients have been divided into Groups A (AD) and B (non-AD). However, patients have also been categorised based on ATN classification, which considers CSF Aβ and pTau concentration. As a result, group A has been subdivided into Group A' and Group A", including, respectively, A+T+N+ and A+ T- N+. A technical reason explains this choice: the current analysis is a part of a more extensive study involving in vivo and in vitro analyses aimed to determine whether and how the CX3CL1 signalling pathway can change during AD onset and progression in terms of transcription factor's expression. The project's final goal is to identify possible targets for novel treatment strategies among the molecules involved in the CX3CL1 pathway, including many transcription factors and metalloproteinases. To reach this goal is mandatory to evaluate microglia activation (or response) to various stimuli, counting Aβ and Tau protein, considering that different stimuli evoke different responses. For instance, Aβ fibrils vs Ab oligomers, or Aβ aggregates vs Tau aggregates, elicit diverse activation mechanisms in microglia, resulting in different phenotypes or subsets of these cells. It is well-known that some phenotypes and mechanisms, namely the "primed" microglia, are strongly associated with the progression of the disease and represents a critical moment within the pathophysiology of AD [Leng F, Edison P. Nat Rev Neurol. 2021;17(3):157-172. doi: 10.1038/s41582-020-00435-y]. In the current study, CSF specimens are requested to show different ATN characteristics with the aim of evaluating possible different responses to be evoked by microglia. Although preliminary, we have reached interesting, almost surprising, results, which are still to be refined for submission. In the current analyses, differences in CX3CL1 levels between A' and A" have been reported. However, the small sample size and study set of the current study make it hazardous to try to explain them through any hypotheses. Enlarging the current sample size and analysing the in vitro and in vivo results will help draw likely conclusions.
3) The reported statistics of the results is not coherent. From the methods (2.4. Statistical Analyses.),
"While the variance found between the different samples as a function of the analyzed variables was determined according to one-way ANOVA (p* value). Furthermore, through the distribution according to student' T, the different samples for the different variables were compared about the significance of the data obtained (p-value)."
I am sorry to say, but the above is not coherent. Am I suppose to assume that it is a one-way ANOVA test (for those with Group A', Group A'' and Group B) and the P-values were reported (which is labeled as p* value)? If so, why is the Aβ 42/40 ratio P-value result in Figure 1 shown as 1.61? A P-value as commonly understood should not have values above 1. The manuscript says that p* value is 'the variance of the values', which is not the case as the vast majority of the time, σ2 is the variance, not p*.
3) Concerning the p* value reported in Table 1 for the ratio Aβ 42/40, it was a typing error; the effective one is 0.03 (the new version has been correct). The evaluation made is determined according to a one-way ANOVA for which the different groups were compared with the total number of patients analyzed.
4) Figure 1 as an image is not coherently depicted either in the manuscript.
The above is a screen capture of what I see and it seems that the figure was badly resized and also flipped. There are also no legends detailing which one is Group A and which one is Group B. The caption text, "All groups were valuate in significance about T of student shown a p < 0.001." is also incoherent. Is this suppose to be a Student's T-test?
4) We are sorry for this inconvenience (unwanted rotation of the image in Fig.1), which we had reported to the editor by sending a new format in which the figure had been fixed. Indeed, the version sent by the publisher for review appears in the correct configuration.
Relatively the considerations related to the student t-test have been eliminated, as evidenced incongruent and on the suggestion of colleagues expert in statistics the analyzes were conducted according to Mann Whitney R - test and included in the revised work.
5) In Figure 5, I am guessing that the P-values were depicted on the Figure based on performing a Student's T-test between the 2 of the Groups, i.e. Group A' vs Group A'', Group A' vs Group B, Group A'' vs Group B. However, the P-values does not seem to match the box-plots. I would assume the Group A' vs Group B and Group A'' vs Group B comparisons should have a more significant P-value than the Group A' vs Group A'' comparison. From just looking at the box-plots, it is hard for me to believe that the P-value of Group A' vs Group A'' is significant at P<0.001.
5) As already said before, we agree with the inconsistency consideration regarding the data reported. And we developed, as mentioned, the statistical analyzes according to the Mann Whitney R -test, given the limited number of values available (<30 per group) relying on a non-parametric test, and we removed the values relating to Student T-test, which were reported.
Overall, I think there is worthy data in the manuscript to report. However, I would suggest that the authors edit the manuscript extensively to better present the data, analyses and results.
Round 2
Reviewer 1 Report
Authors addressed all my comments and suggestions.
I recommend acceptance of this manuscript for publication.
Author Response
We thank you for her/his review.
Reviewer 2 Report
The authors have made extensive changes to the manuscript and it reads much better. However, there are still some minor points,
1) The authors mentioned they added the following text,
"In the aim to measure CX3CL1 in AD patients and compare it to a non-AD control group, patients have been divided into Groups A (AD) and B (non-AD). However, patients have also been categorised based on ATN classification, which considers CSF Aβ and pTau concentration. As a result, group A has been subdivided into Group A' and Group A", including, respectively, A+T+N+ and A+ T- N+. A technical reason explains this choice: the current analysis is a part of a more extensive study involving in vivo and in vitro analyses aimed to determine whether and how the CX3CL1 signalling pathway can change during AD onset and progression in terms of transcription factor's expression. The project's final goal is to identify possible targets for novel treatment strategies among the molecules involved in the CX3CL1 pathway, including many transcription factors and metalloproteinases. To reach this goal is mandatory to evaluate microglia activation (or response) to various stimuli, counting Aβ and Tau protein, considering that different stimuli evoke different responses. For instance, Aβ fibrils vs Ab oligomers, or Aβ aggregates vs Tau aggregates, elicit diverse activation mechanisms in microglia, resulting in different phenotypes or subsets of these cells. It is well-known that some phenotypes and mechanisms, namely the "primed" microglia, are strongly associated with the progression of the disease and represents a critical moment within the pathophysiology of AD [Leng F, Edison P. Nat Rev Neurol. 2021;17(3):157-172. doi: 10.1038/s41582-020-00435-y]. In the current study, CSF specimens are requested to show different ATN characteristics with the aim of evaluating possible different responses to be evoked by microglia. Although preliminary, we have reached interesting, almost surprising, results, which are still to be refined for submission. In the current analyses, differences in CX3CL1 levels between A' and A" have been reported. However, the small sample size and study set of the current study make it hazardous to try to explain them through any hypotheses. Enlarging the current sample size and analysing the in vitro and in vivo results will help draw likely conclusions."
However, I don't see it in the new revise document, even though I see the extensive changes made. Please check to see if this very important description of the A' and A'' groupings based on ATN criteria is included in the final manuscript.
2) As for Figure 1, while it has been rotated, I am still seeing it flipped. See below,
Even if it is not flipped, the authors should also ensure the sizing is correct. The image look stretched and there should be a legend to describe the shaded box-plots.
3) I am still concerned about the language use for p*,
Is it correct or should it be the regular understanding of p-values? If it is variance, shouldn't the symbol be σ2 instead?
Author Response
1) The authors mentioned they added the following text,
"In the aim to measure CX3CL1 in AD patients and compare it to a non-AD control group, patients have been divided into Groups A (AD) and B (non-AD). However, patients have also been categorised based on ATN classification, which considers CSF Aβ and pTau concentration. As a result, group A has been subdivided into Group A' and Group A", including, respectively, A+T+N+ and A+ T- N+. A technical reason explains this choice: the current analysis is a part of a more extensive study involving in vivo and in vitro analyses aimed to determine whether and how the CX3CL1 signalling pathway can change during AD onset and progression in terms of transcription factor's expression. The project's final goal is to identify possible targets for novel treatment strategies among the molecules involved in the CX3CL1 pathway, including many transcription factors and metalloproteinases. To reach this goal is mandatory to evaluate microglia activation (or response) to various stimuli, counting Aβ and Tau protein, considering that different stimuli evoke different responses. For instance, Aβ fibrils vs Ab oligomers, or Aβ aggregates vs Tau aggregates, elicit diverse activation mechanisms in microglia, resulting in different phenotypes or subsets of these cells. It is well-known that some phenotypes and mechanisms, namely the "primed" microglia, are strongly associated with the progression of the disease and represents a critical moment within the pathophysiology of AD [Leng F, Edison P. Nat Rev Neurol. 2021;17(3):157-172. doi: 10.1038/s41582-020-00435-y]. In the current study, CSF specimens are requested to show different ATN characteristics with the aim of evaluating possible different responses to be evoked by microglia. Although preliminary, we have reached interesting, almost surprising, results, which are still to be refined for submission. In the current analyses, differences in CX3CL1 levels between A' and A" have been reported. However, the small sample size and study set of the current study make it hazardous to try to explain them through any hypotheses. Enlarging the current sample size and analysing the in vitro and in vivo results will help draw likely conclusions."
However, I don't see it in the new revise document, even though I see the extensive changes made. Please check to see if this very important description of the A' and A'' groupings based on ATN criteria is included in the final manuscript.
The specifications were incorporated into the results and discussion
2) As for Figure 1, while it has been rotated, I am still seeing it flipped. See below,
Even if it is not flipped, the authors should also ensure the sizing is correct. The image look stretched and there should be a legend to describe the shaded box-plots.
The shades of gray for the different box-plots have been inserted in all the figures.
3) I am still concerned about the language use for p*,
Is it correct or should it be the regular understanding of p-values? If it is variance, shouldn't the symbol be σ2 instead?
p-value was replaced with σ2.